# In Vitro Microevolution and Co-Selection Assessment of Florfenicol Impact on *Escherichia coli* Resistance Development

**DOI:** 10.3390/antibiotics12121728

**Published:** 2023-12-14

**Authors:** Ádám Kerek, Bence Török, Levente Laczkó, Gábor Kardos, Krisztián Bányai, Zoltán Somogyi, Eszter Kaszab, Krisztina Bali, Ákos Jerzsele

**Affiliations:** 1Department of Pharmacology and Toxicology, University of Veterinary Medicine Budapest, 1078 Budapest, Hungary; bencetorok99@gmail.com (B.T.); banyai.krisztian@univet.hu (K.B.); somogyi.zoltan@univet.hu (Z.S.); jerzsele.akos@univet.hu (Á.J.); 2National Laboratory of Infectious Animal Diseases, Antimicrobial Resistance, Veterinary Public Health and Food Chain Safety, University of Veterinary Medicine Budapest, 1078 Budapest, Hungary; kg@med.unideb.hu (G.K.); kaszab.eszter@univet.hu (E.K.); 3Institute of Metagenomics, University of Debrecen, 4032 Debrecen, Hungary; nagyonlevente@gmail.com; 4Veterinary Medical Research Institute, 1143 Budapest, Hungary; bali.krisztina@univet.hu

**Keywords:** microevolution, co-selection, MEGA-plate, *Escherichia coli*, florfenicol, NGS

## Abstract

The issue of antimicrobial resistance is becoming an increasingly serious challenge in both human and veterinary medicine. Prudent antimicrobial use in veterinary medicine is warranted and supported by international guidelines, with the Antimicrobial Advice Ad Hoc Expert Group (AMEG) placing particular emphasis on the critically important group B antimicrobials. These antimicrobials are commonly employed, especially in the poultry and swine industry. The impact of florfenicol, a veterinary antibiotic, was studied on the resistance development of *Escherichia coli*. The aim of the study was to investigate the effect of the use of florfenicol on the development of phenotypic and genomic resistances, not only to the drug itself but also to other drugs. The minimum inhibitory concentrations (MICs) of the antibiotics were investigated at 1×, 10×, 100× and 1000× concentrations using the adapted Microbial Evolution and Growth Arena (MEGA-plate) method. The results demonstrate that florfenicol can select for resistance to fluoroquinolone antibiotics (167× MIC value increase) and cephalosporins (67× MIC value increase). A total of 44 antimicrobial resistance genes were identified, the majority of which were consistent across the samples. Chromosomal point mutations, including alterations in resistance-associated and regulatory genes (*acrB*, *acrR*, *emrR* and *robA*), are thought to trigger multiple drug efflux pump activations, leading to phenotypically increased resistance. The study underscores the impact of florfenicol and its role in the development of antimicrobial resistance, particularly concerning fluoroquinolone antibiotics and cephalosporins. This study is the first to report florfenicol’s dose-dependent enhancement of other antibiotics’ MICs, linked to mutations in SOS-box genes (*mdtABC-tolC*, *emrAB-tolC* and *acrAB*-*tolC*) and increased multidrug efflux pump genes. Mutations in the regulatory genes *acrR*, *emrR* and *rpbA* support the possibility of increased gene expression. The results are crucial for understanding antimicrobial resistance and its development, highlighting the promising potential of in vitro evolutionary and coselection studies for future research.

## 1. Introduction

Antimicrobial resistance is currently one of the most critical issues in both human and animal health. Conservative estimates already suggest annual human deaths numbering 700,000 and economic losses in the trillions of dollars due to this issue [1], with projected mortality figures reaching 10 million by 2050 [2]. Antibiotics have been a cornerstone in treating infectious diseases since the 1940s, with the discovery of several new classes of antibiotics during the 1960s representing the golden age of antibiotics [3]. Infectious diseases rank as the second leading cause of global mortality, with particular concern arising from panresistant strains of Gram-negative bacteria [4]. Target-based antibiotic development has not yielded significant breakthroughs, necessitating a greater emphasis on target-based antibiotic research [5].

From both animal and human health perspectives, it is crucial to delineate the precise mechanisms of resistance to various groups of active substances and explore potential correlations among them, particularly in terms of coresistance. Florfenicol is a broad-spectrum veterinary antibiotic belonging to the phenicol group, which functions as an inhibitor of bacterial protein synthesis, exerting bacteriostatic activity by targeting the 50S ribosome [6]. Bacteria develop resistance to florfenicol through various mechanisms, ranging from antibiotic modification to efflux pumps. Currently, resistance to florfenicol is on the rise due to global trade and misuse [6].

In our research, we adapted [7] the MEGA-plate method [8], for the first time in veterinary medicine, to investigate the phenotypic and genotypic expressions of induced resistance. Within the categorization established by the Antimicrobial Advice Ad Hoc Expert Group (AMEG)—encompassing categories A, B, C and D—AMEG B antibiotics are recognized as critically important for human medicine. Despite this designation, these antibiotics continue to be extensively utilized in farm animals, significantly contributing to the widespread emergence of antibiotic resistance. Notably, resistance is on the rise in the poultry sector, affecting AMEG B antibiotics such as third and fourth generation cephalosporins and fluoroquinolones. It is crucial to underscore that the use of cephalosporins in poultry is not authorized [9,10], highlighting the potential for the concurrent development of coresistance. Considering the AMEG B category, it is not advisable to prioritize substances within this category as the primary choice in veterinary medicine. When selecting a treatment, it is essential to conduct sensitivity testing for an accurate determination of the appropriate substance. Based on the results, a judicious approach involves favoring substances from the less critical AMEG D or AMEG C categories. This strategy aligns with the objective of promoting responsible and sustainable use of antimicrobial agents in veterinary practices [11].

This system allows for the migration and adaptation of the bacteria under study in a large space, which is also a structured environment. The movement of the bacteria, subjected to continuous selection pressure, is enabled by the uppermost soft agar layer, which spreads by chemotaxis as it consumes nutrients. The size of the plate allows for the formation of mutations during the numerous generation changes and maintains the gradient of the antibiotic under study despite diffusion [8,12]. It is, however, a slower process than adaptation in a microenvironment [13], which can be explained by the additional time of 10–12 days, on average, required for the plate to fully grow. Furthermore, a limitation of the method may be that among the development of several parallel lineages, it is not certain that the frontline is driven by those that are best adapted [8]. The method is a population genetics study describable by Muller plots [14]. Our objective was to examine the impact of florfenicol on the development of coresistance to specific active substances approved for poultry use, with particular emphasis on cephalosporins not authorized for poultry use, through evolutionary and co-selection studies.

## 2. Results

### 2.1. MIC Value Change

Within the MEGA-plate, bacteria reached the 1000× concentration level within 10 days. The MIC values were, subsequently, determined for the samples collected from each compartment with varying antibiotic concentrations. As demonstrated, the MIC values of the most active substances increased as early as within the 1× concentration compartment and continued to rise with increasing concentrations, particularly at the 1000× level. Two exceptions were observed, colistin and neomycin, whose MIC values remained unchanged even at 1000× the concentration of florfenicol (Table 1).

### 2.2. Extended-Spectrum Beta-Lactamase (ESBL) Production Screening

None of the samples exhibited a reduction of at least three-fold in MIC when combined with clavulanic acid compared to the standalone beta-lactam antibiotics, both ceftazidime and cefotaxime. Consequently, it can be concluded that clinically significant ESBL production was not detected phenotypically, which is further supported by our genomic studies (Table 2).

### 2.3. Sequencing Data Quality

The results from QUAST (v5.2.) software [15], used for quality control of the contigs, are shown in Table 3.

GenomeScope profiles were created for each sample to assess the general genome characteristics. These features are crucial for genome evolution studies and aid in choosing parameters for subsequent analyses. The kmer linear plot (Figure 1) after error correction offers estimations of coverage, genome size and kmer size. These qualitative data ensure that the sequencing was of sufficient quality and that the contigs were suitable for bioinformatic analysis. The frequency histograms of the kmer are consistent with *Escherichia coli* (*E. coli*).

Checkm (v1.1.6.) and Kraken Software (v1.1.1.) identified 100% identity with the *E. coli* bacterial species in all samples. To assess the genomic diversity among the genomes, we utilized ANI software (v2.0.), conducting a taxonomic analysis of the genomes from various phylogenetic lineages. The resistant lines selected for analysis, which had undergone four drug concentration steps from the initial 0× to the 1000× sample treated with the highest drug concentration, did not exhibit differences in the average nucleotide identity of orthologous genes shared between the two genomes. These two samples displayed an ANI of 100% (Figure 2), meeting the criterion of ≥95% ANI for same-species identity [16].

### 2.4. Antimicrobial Resistance Gene (ARG) Set

Regarding the antimicrobial resistance gene (ARG) set, our identified ARGs, meeting the strict threshold criteria of the CARD database, exhibited a coverage percentage and sequence identity percentage exceeding 90%. These ARGs were consistent across all samples, totaling 44 identified ARGs capable of conferring resistance to a range of 22 antibiotics, disinfectants and various dyes (Appendix A). Notably, the *ampC* and *ampH* genes, which provide resistance to beta-lactam (penam and cephalosporin) antibiotics through enzymatic inactivation, were present. The *ampH* gene was identified as a mobile genetic element (MGE) in the 1× and 10× samples. Additionally, the presence of the *bacA* gene, responsible for conferring peptide antibiotic resistance through target alteration, was noteworthy, as it was identified as an MGE and phage-encoded in all samples. In the 10× samples, the *emrB* gene, associated with fluoroquinolone resistance, was found to be phage-encoded. Several identified multidrug efflux pump genes may contribute to resistance development against phenicols (Table 4).

A total of 44 ARGs were identified, classified by drug group and by resistance mechanism (Figure 3). The majority of these ARGs are associated with resistance to fluoroquinolones, totaling 21, out of which 19 genes encode efflux pumps. The second most prevalent category involves genes that confer resistance to penicillins, comprising 17 genes in total. Among these, the majority (14 genes) are associated with efflux pumps, while two genes are responsible for enzymatic inactivation (*ampC* and *ampH*). Notably, genes related to target alteration mechanisms (four genes) were exclusively identified in the context of peptide antibiotics.

The analysis conducted with the mlplasmid v2.1. software revealed that all identified antimicrobial resistance genes were encoded within the chromosomes. Additionally, the use of VirSorter v2.2.2. software identified *bacA* and *emrB* as phage-encoded genes. Furthermore, when employing the MobileElementFinder v1.0.3. software, both the *ampH* and *bacA* genes were recognized as MGEs (Appendix A).

### 2.5. Serotyping and Changes in Virulence Factors

The sequencing data provided us with the ability to determine the serotype of the studied strain. Consequently, we identified specific polysaccharides for the O6 serotype (*wzx*, *wzy*), as well as protein-based antigens H1 and H12 (*fliC*). To investigate whether varying concentrations of florfenicol had an impact on the number of virulence factors, we identified a total of 40 identical virulence factors present in each sample. Interestingly, there were only two genes that differed among the samples. The gad gene (protein function: glutamate decarboxylase) was detected in both the 1× and 100× samples, while the hha gene (protein function: hemolysin expression modulator) was only found in the 1× sample. Overall, it appears that the active substance florfenicol did not significantly influence the presence of these virulence factors (Appendix A).

### 2.6. Mutations

A total of 8753 mutations were detected in the samples, with 4623 of them having an identified function. The distribution of total mutations varied between 1722 and 1777 mutations per sample, while the distribution of identified mutations ranged from 918 to 933 mutations per sample. When comparing these mutations to the baseline 0× sample, there was an overlap of 101.3% for the 1× sample, 101.1% for the 10× sample, 99.9% for the 100× sample and 99.7% for the 1000× sample. The distribution of each mutation type in the samples can be found in Table 5. All mutations are interpreted relative to the SYNB8802 strain used as the reference strain for bioinformatic analysis.

The majority of mutations observed in the samples were of the single-nucleotide polymorphism (SNP) type, with the highest number identified in the 1× sample. Following SNP mutations, complex mutations involving complete amino acid substitutions were the next most common, with a significant presence in the 1× sample as well. Deletion mutations, resulting in the loss of a single amino acid, were most frequently observed in the 10× and 100× samples, while insertion mutations had the lowest number of identified occurrences (Appendix A).

When examining mutations relevant to antimicrobial resistance, genomic changes attributed to SNPs were found to potentially explain the increased MIC values against several antibiotics following exposure to florfenicol, as outlined in Table 6. In addition, we observed in the case of the *marR* gene that concentrations of florfenicol 10× and 100× induced a deletion in the gene, which triggers its activation as a repressor, thereby activating the multidrug efflux pump *acrAB*.

Comparing our samples to the reference strain, we identified mutations in the *mdtC* and *mdtN* genes present in all samples. These mutations could potentially explain the inherently high MIC values for certain drugs due to the presence of active multidrug efflux pumps. Additionally, mutations in the *emrR* gene, involved in regulating multidrug efflux pumps, were detected in the 10× and 100× samples. Mutations in the *acrB* gene, which determines the function of the *acrAB-tolC* pump system, as well as in the *acrR* and *robA* genes, which regulate this system, could result in increased pump function.

The *mdtC* gene encodes a transporter that forms a heteromultimeric complex with the *mdtB* gene, creating a multidrug transporter. The *mdtBC* complex is part of the *mdtABC-tolC* efflux system. In the absence of *mdtB*, *mdtC* can form a homomultimeric complex, leading to a functional efflux system with narrower drug specificity [17]. We detected all three of these genes in our samples, suggesting that the mutation inherited from the reference genome contributed to the high initial MICs of the *E. coli* strain against several drugs. The *mdtN* gene encodes a multidrug resistance efflux pump that may play a role in resistance to puromycin-acriflavine and tetraphenylarsonium chloride [18,19]. It was also present in all samples, indicating that this mutation was inherently present in our strain compared to the reference strain, contributing to the high initial MIC values for some drugs.

The *emrR* gene acts as a negative regulator of the *emrAB-tolC* multidrug efflux pump complex in *E. coli*. Mutations in this gene result in the expression of the *emrAB-tolC* complex [20]. In our samples, mutations observed in the 10× and 100× samples might have led to the expression of the multidrug efflux pump. All components of this complex pump system, including *emrA*, *emrB*, and *tolC* genes, were detected in all samples. The *emrA* and *emrB* genes are primarily responsible for encoding the efflux pump involved in pumping out enrofloxacin, while the *tolC* gene encodes the efflux pump responsible for expelling almost all other drugs. This could potentially explain the increase in MIC values for other drugs in response to high concentrations of florfenicol.

The *acrB* gene is a component of the *acrAB-tolC* multidrug efflux complex protein. AcrB functions as a heterotrimer, comprising the inner membrane component, and plays a crucial role in substrate recognition and energy transduction by operating as a drug-proton antiporter [21,22,23,24]. Mutations in the *acrB* gene were induced by 100× and 1000× concentrations of florfenicol, which may also contribute to the increased MIC values. The *acrR* gene functions as a repressor of the *acrAB-tolC* multidrug efflux complex. Mutations in *acrR* can lead to high levels of antibiotic resistance [25]. Mutations in this gene were only detected in samples treated with the 1000× florfenicol concentration, which likely contributed to the additional increase in the MIC.

The *robA* gene acts as a positive regulator of the genes encoding the *acrAB* efflux pump and shares structural similarities with the *soxS* and *marA* genes [26]. Mutations in this gene were also only detected in the sample treated with the 1000× florfenicol concentration and may have contributed to the elevated MIC values.

## 3. Discussion

In this study, we employed the modified MEGA-plate method [7] to investigate the development of antibiotic resistance in *E. coli* strains under selection pressure by veterinary antibiotic florfenicol. The escalating resistance to florfenicol, a commonly used antibiotic in the swine industry, is concerning, with resistance levels rising from 2.1% in 2004 to 18.1% in 2017 [27], Additionally, in broiler chickens, florfenicol administration during rearing has been linked to a significant increase in *E. coli* resistance to phenicols [28].

Our findings reveal a phenotypic decrease in florfenicol susceptibility, attributed to genome mutations mediating the upregulation of efflux pump expression. These genome mutations are a consequence of stress-induced changes in the bacterium, which contribute to the expression of resistance genes. Additionally, it is possible that clones with active efflux pumps were selectively favored during this process. Langsrud et al. previously reported a 1.5–20× increase in MIC for antibiotics due to stress-induced cross-resistance to benzalkonium chloride [29]. In our study, we observed a 2–167× increase in MIC values against various antibiotics following treatment with 1000× concentrations of florfenicol. Cheng et al. also demonstrated an increased MIC against tigecycline in *Acinetobacter baumani* strain [30]. Ding et al. were able to achieve similarly elevated MIC values in *Streptoccus* strain by passaging [31]. The antimicrobial agents were categorized based on their public health significance by AMEG of European Medicines Agency (EMA). Notably, florfenicol caused a substantial MIC increase to a veterinary fluoroquinolone, enrofloxacin, classified as an AMEG B [32], thus highlighting the importance of cross-resistance development to critically important antimicrobials, although florfenicol itself is an AMEG C substance. Chueca et al. exposed *E. coli* strains to essential oils for 10 days, leading to increased MIC values against several antibiotics [33]. The exposure of bacteria to antibiotics at subinhibitory concentrations can induce an SOS stress response, potentially causing a transient elevation in mutation rates [34,35]. Notably, no prior studies have reported such am extensive cross-resistance development across different antibiotic classes following exposure to varying doses of florfenicol. Resistance to penicillins and cephalosporins is presumed to be linked to efflux pump activation.

Among the 44 identified ARGs, the presence of genes responsible for enzymatic inactivation, particularly *ampC* and *ampH*, is of particular concern. These genes are associated with the overproduction of beta-lactamase enzymes, which can induce resistance to cephalosporins. Resistance to critically important agents includes the presence of genes, such as *bacA*, *pmtF*, *eptA* and *ugd*, which can confer resistance through target modification against colistin. We identified the presence of the *marA* gene, which can induce multidrug resistance through reduced permeability. Additionally, we identified two genes responsible for developing resistance to drug classes (AMEG A) reserved for human healthcare (*marA* and *tolC*). Furthermore, 59.1% of all identified genes were found to be responsible for resistance to essential drug classes for both public health and animal health (*acrA*, *acrB*, *acrE*, *acrF*, *acrS*, *ampC*, *ampH*, *bacA*, *CRP*, *emrA*, *emrB*, *emrR*, *eptA*, *evgA*, *evgS*, *gadW*, *H-NS*, *marA*, *mdtE*, *mdtF*, *mdtH*, *mdtM*, *pmrF*, *tolC*, *ugd* and *cojI*). Notably, the *ampH* and *bacA* genes were encoded as MGEs, potentially enabling the transfer of these crucial ARGs to other bacterial strains. Surprisingly, we did not detect the *floR* gene, responsible for florfenicol resistance in *Salmonella enterica*, in our *E. coli* samples [36].

The activation of multidrug efflux pumps appears to underlie the elevated MIC values against most tested drugs. Previous research by Ma et al. elucidated the responsiveness of the *acrAB* gene-encoded complex pump system to various environmental stresses [37], In our study, we observed deletion mutations in the *marR* gene for the 10× and 100× florfenicol samples, and it is well-documented that mutations in the *marR* gene can upregulate the *acrAB* operon [37], a key transcriptional regulator of multiple antibiotic resistance genes [38]. Additionally, Pourahmad et al. reported a reduced antibiotic efficacy in *E. coli* due to the overactivation of the *acrAB-tolC* efflux pump complex [39]. Yaqoob et al. described the *acrR* and *robA* genes as regulators of this pump system [40], and we identified mutations in these genes in the 1000× sample. Interestingly, when synthetic antibiotics, honey, and various plant alkaloids were applied, they enhanced pump system function and reduced oxidative stress in an *E. coli* strain [39]. Maslowska et al. identified two pivotal genes, *lexA* and *recA*, governing the SOS stress response, which involves the upregulation of at least 50 additional genes [41]. In our study, we consistently detected both genes in all samples and identified most of the SOS-box genes under their regulation. Notably, single-point mutations in genes encoding multidrug efflux pumps were common, potentially accounting for the observed gene expression.

However, it appears that the action of these multidrug efflux pumps may not be adequate to effectively expel peptide and aminoglycoside antibiotics from *E. coli*. Babosan et al. demonstrated that the presence of *qnr* genes and deletion of the *recB* gene contributed to SOS stress induction by aminoglycosides [42]. In our study, we identified the *recB* gene in our *E. coli* strain but observed no mutations. This might explain the lack of an increase in MIC against neomycin. The minor increase in MIC for beta-lactam antibiotics is likely attributed to efflux pump activity, although we do not suspect the expression of genes responsible for enzymatic inactivation. Such gene expression would have resulted in a much greater increase in MIC values, and we did not detect mutations in these genes.

In summary, our findings suggest that reduced susceptibility to florfenicol and other antimicrobials may be linked to mutations affecting three types of multidrug efflux pump systems encoded by the *mdtABC-tolC*, *emrAB-tolC* and *acrAB-tolC* genes. Furthermore, mutations in the *acrR*, *emrR* and *rpbA* genes that regulate these efflux pumps enhance their functionality. While our study unveils valuable insights into the mechanisms driving antimicrobial resistance in *E. coli*, further research is needed to fully comprehend the intricate interplay between stress-induced gene activation, efflux pump systems and genomic mutations in the context of antibiotic resistance. Our results support the need for future in silico studies targeting the expression of genes associated with established resistance. A deletion of the *marR* gene was shown to occur in response to 10× and 100× concentrations of florfenicol, a deletion that Notka et al. have previously shown to act as a positive regulator of the *acrAB* efflux pump system and to result in a 16× increase in MIC with fluoroquinolones [43]; in our studies, we observed similar MIC increases of an order of magnitude not only for enrofloxacin but also for florfenicol, cephalosporin and oxytetracycline. The point mutations and deletions identified underscore the importance of resistance induced by previously demonstrated repressor genes in a cascade-like SOS stress response, but the association of individual agents in increasing resistance has not been previously investigated in a comprehensive manner like our study. We need to highlight the fact that certain active substances used in veterinary medicine may induce co-selection against other agents, such as cephalosporins, which may be of public health importance. Our results confirmed the value of continuing the study with transcriptomic methods to support the phenotypic expression of genotypically occurring mutations.

In veterinary practice, understanding and managing antibiotic resistance to antibiotics is of paramount importance for the preservation of animal health and the safety of animal populations. Our research has shed light on the fact that antibiotic resistance not only poses a significant threat to animals but can also have profound implications for human health. The spread of antibiotic resistance in the *E. coli* bacterium is of particular concern for both animal and human populations. The results of our studies indicate that the development of antibiotic resistance is closely tied to genetic mutations occurring in the genomes of bacteria exposed to antibiotics. Antibiotic resistance evolves as bacteria undergo genetic changes and activate genes that confer resistance to various antibiotics. The rise in resistance to critically important antibiotics, such as fluoroquinolones, is especially worrisome. These antibiotics are of great importance to human health, and the emergence of resistance poses a serious health risk to human populations.

## 4. Materials and Methods

### 4.1. Tested Bacterial Strain

The tested bacterial strain, *E. coli*, reference strain ATCC 25922 (LGC Ltd., Teddington, UK), originally isolated in Seattle in 1946, was employed for the studies. The selection of the strain for the study was made because it is a reference strain widely used and well documented in the scientific literature. It is particularly suitable for resistance studies as it is sensitive to various antibiotics, allowing for a more precise evaluation of the drug’s effects.

### 4.2. Preparation of the MEGA-Plate

The experiments were conducted in a 60 cm × 30 cm polycarbonate tray, which was constructed from 5 mm thick material (Innoterm Ltd., Budapest, Hungary) and assembled using waterproof tetrahydrofuran glue. The bottom of the tray was divided into nine equal compartments to facilitate the segregation of the media with increasing antibiotic concentrations in the lower layer. To disinfect the tray, we filled it with 7.5% hydrogen peroxide (VWR International Kft., Debrecen, Hungary) [44,45], and the inner surface and rim of the cover plate were wiped with a 1% NaOCl aqueous solution (Merck KGaA, Darmstadt, Germany). Subsequently, it was incubated in a sterile chamber for 15 min, after which the hydrogen peroxide was removed using a vacuum pump, and a 30 min UV light treatment was initiated.

Three layers were established during the medium infusion. The bottom layer consisted of nine discrete compartments, with antibiotic concentrations of interest ranging from 0×, 1×, 10×, 100× to 1000× of the active ingredient, progressing from the edges of the tray inward (compartments 1–5). Layer 2 formed a continuous solid layer that ensured homogeneity between layers 1 and 3. Layer 3 represented the semi-fluid layer, which enabled the diffusive growth in bacteria. Bacterial growth occurred against increasing drug concentration gradients in this layer. For the preparation of the culture medium, we employed BD Bacto Agar (VWR International Ltd., Debrecen, Hungary) at a concentration of 2%, except for the third layer, where we prepared a semi-liquid layer at a concentration of 0.28%. One LB-Lennox (VWR International Ltd., Debrecen, Hungary) capsule per liter of medium was added as a nutrient supplement. In our first preliminary experiments, the bacteria showed no growth inwards from the first concentration line, and we decided to add another capsule to the top layer. Subsequently, the bacteria showed the expected growth inwards as originally conceived. On the basis of our prior experience, an additional capsule was added to the top layer. To prevent fungal contamination, cycloheximide (Merck KGaA, Darmstadt, Germany) was included in the medium at a concentration of 64 µg/mL. In the lower and middle layers, 4 mL/liter of black acrylic stain (Artmie, Budapest, Hungary) was added to provide contrast. The day before the experiment a yellow loop (VWR International Ltd., Debrecen, Hungary) of the *E. coli* strain, which was stored in a Microbank system (VWR International Ltd., Debrecen, Hungary) at −80 °C and used for inoculation, was inoculated into a tryptone soy broth (VWR International Ltd., Debrecen, Hungary) and incubated at 37 °C for 24 h. Finally, the *E. coli* strain was inoculated onto both edges of the plate, which was then placed in a 37 °C incubator (Figure 4).

### 4.3. Antibiotic Susceptibility Testing

To initiate the experiment, we determined the minimum inhibitory concentrations (MICs) of certain antibiotics for the *E. coli* strain. We investigated the susceptibility to ceftriaxone, cefquinome, cefotaxime, ceftiofur, colistin, enrofloxacin, amoxicillin, neomycin, oxytetracycline, florfenicol and potentiated sulphonamide. All antibiotics were sourced from Merck KGaA (Darmstadt, Germany). The assay followed the Clinical Laboratory Standards Institute (CLSI) methodology [46]. To provide an evolutionary advantage for the selection of resistant lines, we considered 0.25× of the initial MIC value of florfenicol (16 µg/mL) as the 1× concentration. Concentrations of 10×, 100× and 1000× of this 1/4 concentration were used to prepare the MEGA-plate. We took samples from each compartment containing antibiotic concentrations and inoculated them onto differentiating and selective ChromoBio^®^ Coliform agar (Biolab Zrt., Budapest, Hungary) to ensure no contamination occurred. We conducted our tests in triplicate, and consistently observed highly similar trends. Values in bold indicate an increase in the MIC (µg/mL) compared to the baseline (Table 1). Bacterial colonies from a colony-forming unit were transferred onto tryptone-soy agar (Biolab Zrt., Budapest, Hungary) and stored in a Microbank™ system (Pro-Lab Diagnostics, Richmond Hill, ON, Canada) at –80 °C until further use. In each case, we collected three replicate samples, and each assay was performed in triplicate. The working plates were then incubated at 37 °C for 18–24 h, and the MIC values were assessed by visual observation in comparison to the positive controls.

### 4.4. Assay for ESBL Production

The ESBL production assay was conducted following the CLSI methodology [46]. In this context, we determined the minimum inhibitory concentration (MIC) of tested bacterial strains for ceftazidime, ceftazidime-clavulanic acid, cefotaxime, and cefotaxime-clavulanic acid. A fixed concentration of 4 µg/mL clavulanic acid was included in each dilution for the clavulanic acid combinations. Subsequently, the plates were incubated in a thermostat at 37 °C for 18–24 h. According to the CLSI guidelines, a strain is classified as ESBL-producing if a minimum three-fold reduction in the MIC value of the antimicrobial agent tested in combination with clavulanic acid is observed.

### 4.5. Next-Generation Sequencing

DNA was isolated from the bacterial suspension using the Quick-DNA Fungal/Bacterial Miniprep Kit, D6005 (Zymo research, Murphy Ave., Irvine, CA, USA) according to the manufacturer’s protocol. Paired-end reads generated from the DNA were determined using an Illumina NextSeq 500 sequencer [47]. The procedure used by Illumina products, which we also used in this study, is a “pair end” technique in which single-stranded DNA strands are anchored with oligonucleotides during bridge amplification, the other strand is inserted and bridged. The reverse strand is removed and the fluorescently labeled linked nucleotides are read during sequencing [48,49]. Nucleotide sequences were determined using next-generation sequencing on an Illumina NextSeq 500 sequencer (Illumina, San Diego, CA, USA) following the protocols described previously [50]. For the reversible terminator sequencing (RTS) method, Illumina^®^ Nextera XT DNA Library Preparation Kit (Illumina, San Diego, CA, USA) and the Nextera XT Index Kit v2 Set B (Illumina, San Diego, CA, USA) were employed to prepare Illumina-specific libraries. The DNA samples were diluted to a final concentration of 0.2 ng/μL in nuclease-free water (Promega, Madison, WI, USA) with a total volume of 2.5 μL. The reaction components were used in reduced volumes. In the tagmentation reaction, 5 μL of Tagment DNA buffer and 2.5 μL of Amplicon Tagment Mix were combined. The samples underwent tagmentation at 55 °C for 6 min, utilizing the GeneAmp PCR System 9700 (Applied Biosystems/Thermo Fisher Scientific, Foster City, CA, USA). Subsequently, the samples were allowed to cool to 10 °C before adding 2.5 μL of Neutralize Tagment buffer. Neutralization was carried out for 5 min at room temperature.

For library amplification, 7.5 μL of Nextera PCR Master Mix and 2.5 μL each of the i5 and i7 index primers were mixed with the tagmented DNA samples. The index primers were integrated into the library DNA via 12 PCR cycles, with each cycle comprising the following steps: 95 °C for 10 s, 55 °C for 30 s, followed by 72 °C for 30 s. After the PCR cycles, the samples were held at 72 °C for 5 min and then cooled to 10 °C. Libraries were, subsequently, purified using the Gel/PCR DNA Fragments Extraction Kit from Geneaid Biotech Ltd. (Taipei, Taiwan). The concentration of the purified libraries was determined, and the libraries were pooled and denatured. The denatured library pool, with a final concentration of 1.8 pM, was loaded onto a NextSeq 500/550 High Output flow cell and sequenced using an Illumina^®^ NextSeq 500 sequencer (Illumina, San Diego, CA, USA).

### 4.6. Bioinformatic Analysis

Quality control of the raw sequences was performed using FastQC v0.11.9 [51] and Fastp v0.23.2-3 [52], and the filtering of sequences of inadequate quality was performed using TrimGalore v0.6.6. The read sequences were aligned into longer sequences (i.e., contigs) using MEGAHIT v1.2.9 [53]. QUAST v5.2. software [15] and Busco v5 software [54] were used to quality control the contigs. GenomeScope v2.2 software [55] was used to estimate the overall genome features. All possible open reading frames (ORFs) were determined from the resulting contigs using Prodigal v2.6.3 [56]. Identification of antimicrobial resistance genes (ARGs) among ORFs was performed using the Resistance Gene Identifier (RGI) v5.1.0 against the CARD database [57]. Only genes that met the STRICT threshold criteria defined by the CARD database and showed at least 90% sequence identity and coverage were considered.

To investigate the potential mobility of the identified resistance genes, we used MobileElementFinder (v1.0.3) [58], a program that predicts mobile genetic elements (MGEs) on contigs. For this purpose, only those ARGs that were within the distance of the longest *E. coli*-specific complex transposon in the database were considered potentially mobile. In addition, the plasmid origin of contigs was investigated using PlasFlow v1.1 [59] software and mlplasmids v2.1 software [60] and the presence of phage genomes on contigs was determined using VirSorter v2.2.2 [61] software. The MGE, plasmid and phage genome results were further filtered for hits within 10,000 base pairs. For species identification, Checkm v1.2.2 software [62] and Kraken v1.1.1 software [63] were used. ResFinder version 4.1 [64,65,66] was used to search for chromosomal point mutations, and Snippy v4.6.0 was used to track polymorphisms in the genome. Ectyper v1.0 was used for serotyping [67], and VirulenceFinder 2.0 was used to track changes in virulence factors [65,68,69]. Genomic diversity analysis among genomes was performed using Average Nucleotide Identity (ANI) v2.0 software for taxonomic analysis of genomes from different phylogenetic lineages [16]. For bioinformatic reference analysis, we used the *E. coli* (SYNB8802 strain) genome GCF_020995495.1 available in the National Center for Biotechnology Information (NCBI) database, which was the closest full RefSeq genome overlap [70].

## 5. Conclusions

Our results show that the development of antimicrobial resistance is closely linked to the activation of bacterial multidrug efflux pumps. Mutations in genes that determine these pumps may increase antibiotic resistance, which may explain the observed increase in MIC.

Stress factors (such as antibiotic use) may trigger SOS mechanisms that lead to mutations in the genome. Our study supports the hypothesis that the presence of florfenicol induces significant gene expression through mutations, the resulting multidrug resistance is mainly due to efflux pumps, and the expression of these pumps is affected by a single nucleic acid substitution. On the basis of these results, it may be particularly important to consider the administration of antioxidants alongside antibiotics, which may attenuate SOS processes. As a more far-reaching hypothesis, which needs to be tested, these processes may contribute not only to the spread of strains’ resistance to first-line agents but may also lead to the emergence of resistance to second-line agents, and, therefore, in this context, when selecting a second-line agent following unsuccessful treatment, it may be worthwhile to be careful to choose an agent to which sensitivity may still be maintained.

Understanding and monitoring antibiotic resistance to antibiotics used in veterinary practice is key to protecting animal health and, through this, human health. Our results can help to improve the use of antibiotics and contribute to preventing the spread of antibiotic resistance in both animal and human populations. The intertwined role of veterinary medicine and human health underlines the global importance of science and health in safeguarding our well-being.

## Figures and Tables

**Figure 1 antibiotics-12-01728-f001:**
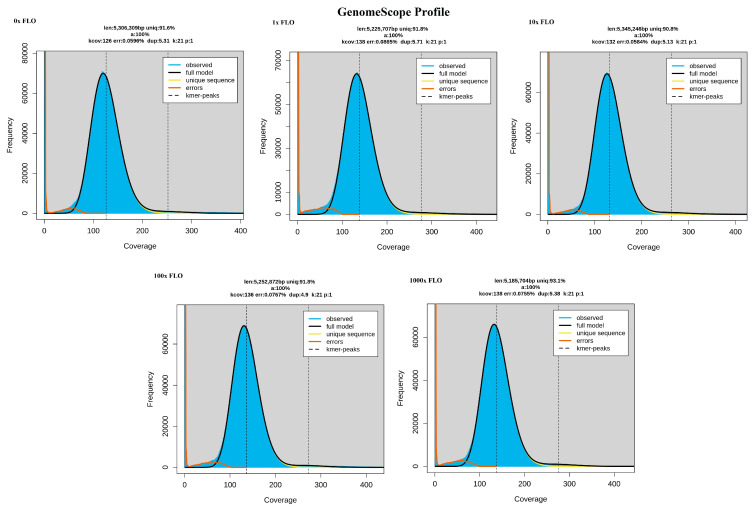
GenoScope profile of each sample. The fit of the model (black line) to the observed kmer frequencies (blue area) is shown for *E. coli* samples taken after 0×, 1×, 10×, 100× and 1000× florfenicol treatments. The genome size, heterozygote ratio and repetitive content of unprocessed short reads can be read.

**Figure 2 antibiotics-12-01728-f002:**
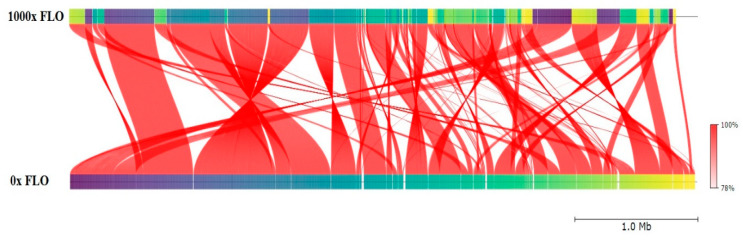
Investigating genetic diversity among genomes. The average nucleotide identity (ANI) of the shared orthologous genes among samples treated with 0× and 1000× florfenicol (FLO) drug concentrations was close to 100%.

**Figure 3 antibiotics-12-01728-f003:**
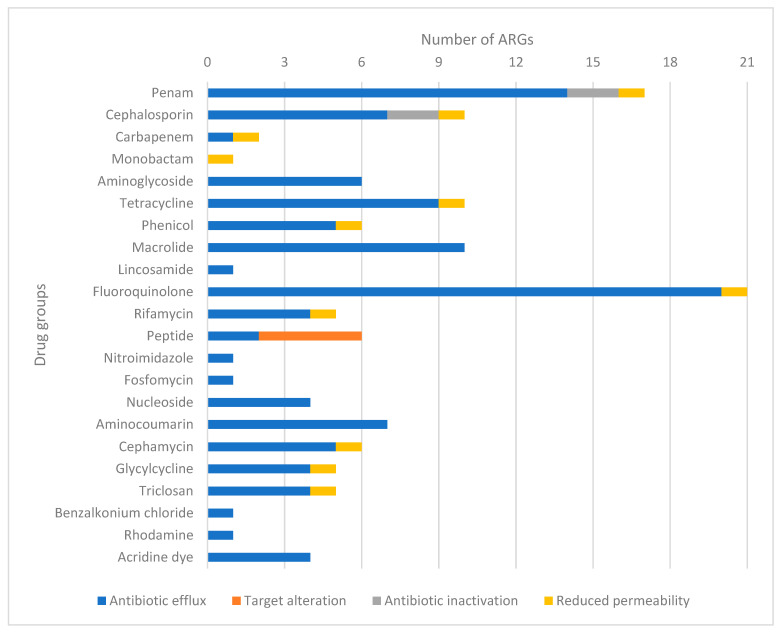
Groupings of each identified antimicrobial resistance genes (ARGs) by drug group and resistance mechanism. The most common mechanism of resistance was conferred by genes encoding efflux pumps against fluoroquinolones. ARGs—antimicrobial resistance genes.

**Figure 4 antibiotics-12-01728-f004:**
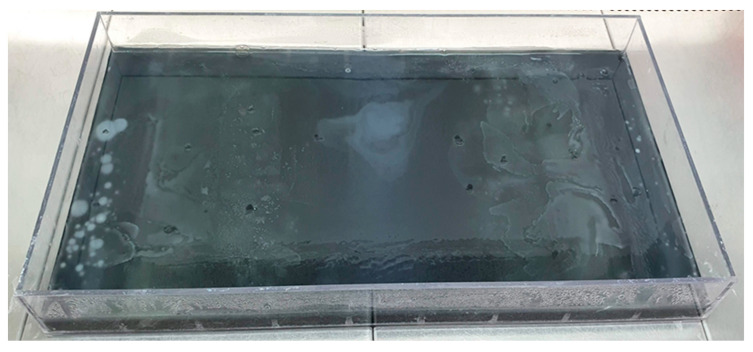
MEGA-plate overgrown with *E. coli* bacteria during 10 days of incubation against increasing florfenicol concentrations.

**Table 1 antibiotics-12-01728-t001:** Effects of increasing concentrations of florfenicol on the MIC of the test compounds. The increase in MIC values of most drugs is due to an adaptation process induced by the presence of the florfenicol, which is explained by the activation of multidrug efflux pumps (described later). However, activated efflux pumps are not able to pump out polymixins and aminoglycosides.

Sample	FLO	ENR	COL	CTX	OTC	PSA	AMX	NEO	CFR	CFT	CFQ
µg/mL											
0 FLO	16	0.003	0.5	0.03	2	8	8	16	0.25	0.06	0.06
1× FLO	**64**	**0.015**	2	**0.125**	**4**	**16**	8	16	**1**	**0.125**	**0.25**
10× FLO	**128**	**0.06**	1	**0.5**	**16**	**32**	8	32	**1**	**0.25**	**0.5**
100× FLO	**>512**	**0.03**	0.125	**0.5**	**32**	**32**	**16**	16	**0.5**	**0.5**	**0.5**
1000× FLO	**>512**	**0.5**	0.5	**2**	**64**	**64**	**32**	16	**0.5**	**2**	**0.5**

FLO—florfenicol, ENR—enrofloxacin, COL—colistin, CTX—cefotaxime, OTC—oxytetracycline, PSA—potent sulphonamide (sulfamethoxazole, trimethoprim), AMX—amoxicillin, NEO—neomycin, CFR—ceftriaxone, CFT—ceftiofur and CFQ—cefquinome. Values in bold indicate an increase in florfenicol-induced MIC.

**Table 2 antibiotics-12-01728-t002:** Results of the ESBL detection study with the Clinical Laboratory Standards Institute (CLSI) recommended combination of ceftazidime (CTZ), cefotaxime (CTX) and clavulanic acid (CLA).

Sample	CTZ	CTZ + CLA	Difference	CTX	CTX + CLA	Difference
	(µg/mL)			(µg/mL)		
0 FLO	0.03	0.015	2×	0.03	0.03	1×
1× FLO	0.03	0.015	2×	0.03	0.03	1×
10× FLO	0.06	0.03	2×	0.06	0.03	2×
100× FLO	0.25	0.125	2×	0.06	0.06	1×
1000× FLO	0.25	0.125	2×	0.125	0.06	2×

FLO—florfenicol, CTZ—ceftazidime, CTZ + CLA—ceftazidime clavulanic acid, CTX—cefotaxime and CTX + CLA—cefotaxime clavulanic acid.

**Table 3 antibiotics-12-01728-t003:** Quality data of the contigs based on the QUAST (v5.2.) software analysis.

Strain	No. of Trimmed Reads	No. of Contigs	Coverage	N50	N75	L50	L75
0× FLO	3,815,950	149	156.64	184,825	101,461	10	19
1× FLO	4,411,308	112	169.864	287,102	132,336	6	13
10× FLO	3,049,718	128	164.932	194,885	104,806	8	17
100× FLO	3,907,460	118	167.917	194,926	107,365	8	17
1000× FLO	3,414,732	119	168.079	184,922	106,685	8	18

FLO—florfenicol.

**Table 4 antibiotics-12-01728-t004:** The set of 44 ARGs identified in the next-generation sequencing was the same for all samples.

Gene	Coverage, %	Identity, %	Mechanism	Resistance
*acrA*	100.00	99.16	antibiotic efflux	cephalosporin, fluoroquinolone, glycylcycline, penam, phenicol, rifamycin, tetracycline and triclosan
*acrB*	100.00	98.64	antibiotic efflux	
*acrD*	100.00	98.04	antibiotic efflux	aminoglycoside
*acrE*	100.00	98.79	antibiotic efflux	cephalosporin, cephamycin, fluoroquinolone and penam
*acrF*	100.00	96.49	antibiotic efflux	
*acrS*	100.00	98.34	antibiotic efflux	cephalosporin, cephamycin, fluoroquinolone, glycylcycline, penam, phenicol, rifamycin, tetracycline and triclosan
*ampC*	100.00	98.15	antibiotic inactivation	cephalosporin and penam
*ampH **	100.00	97.50	antibiotic inactivation	
*bacA ***	99.76	98.17	target alteration	peptide
*baeR*	99.86	96.81	antibiotic efflux	aminocoumarin andaminoglycoside
*baeS*	100.00	90.53	antibiotic efflux	
*cpxA*	100.00	98.47	antibiotic efflux	
*CRP*	100.00	99.21	antibiotic efflux	fluoroquinolone, macrolide and penam
*emrA*	100.00	98.21	antibiotic efflux	fluoroquinolone
*emrB ****	100.00	96.95	antibiotic efflux	
*emrE*	100.00	92.19	antibiotic efflux	macrolide
*emrK*	100.00	97.73	antibiotic efflux	tetracycline
*emrR*	100.00	98.68	antibiotic efflux	fluoroquinolone
*emrY*	100.00	97.73	antibiotic efflux	tetracycline
*eptA*	100.00	91.85	target alteration	peptide
*evgA*	100.00	99.02	antibiotic efflux	fluoroquinolone, macrolide, penam and tetracycline
*evgS*	100.00	96.19	antibiotic efflux	
*gadW*	100.00	99.86	antibiotic efflux	fluoroquinolone, macrolide and penam
*gadX*	100.00	93.82	antibiotic efflux	
*H-NS*	100.00	99.28	antibiotic efflux	cephalosporin, cephamycin, fluoroquinolone, macrolide, penam and tetracycline
*kdpE*	99.26	95.84	antibiotic efflux	aminoglycoside
*marA*	100.00	98.70	reduced permeability	carbapenem, cephalosporin, cephamycin, fluoroquinolone, glycylcycline, monobactam, penam, penem, phenicol, rifamycin, tetracycline and triclosan
*mdfA*	100.00	96.59	antibiotic efflux	benzalkonium chloride, rhodamine and tetracycline
*mdtA*	100.00	95.11	antibiotic efflux	aminocoumarin
*mdtB*	100.00	96.29	antibiotic efflux	
*mdtC*	100.00	94.15	antibiotic efflux	
*mdtE*	100.00	98.62	antibiotic efflux	fluoroquinolone, macrolide and penam
*mdtF*	100.00	97.33	antibiotic efflux	
*mdtG*	100.00	98.21	antibiotic efflux	fosfomycin
*mdtH*	100.00	98.26	antibiotic efflux	fluoroquinolone
*mdtM*	100.00	95.05	antibiotic efflux	acridine dye, fluoroquinolone, lincosamide, nucleoside and phenicol
*mdtN*	100.00	95.64	antibiotic efflux	acridine dye and nucleoside
*mdtO*	100.00	97.08	antibiotic efflux	
*mdtP*	100.00	97.61	antibiotic efflux	
*msbA*	100.00	98.06	antibiotic efflux	nitroimidazole
*pmrF*	100.00	97.63	target alteration	peptide
*tolC*	100.00	97.98	antibiotic efflux	aminocoumarin, aminoglycoside, carbapenem, cephalosporin, cephamycin, fluoroquinolone, glycylcycline, macrolide, penam, penem, peptide, phenicol, rifamycin, tetracycline and triclosan
*ugd*	100.00	96.92	target alteration	peptide
*yoiI*	100.00	98.05	antibiotic efflux	

* MGE in 1× FLO and 10× FLO samples; ** MGE in all samples and phage-encoded; *** phage-encoded in 10× FLO sample; MGEs—mobile genetic elements.

**Table 5 antibiotics-12-01728-t005:** The total number of mutations observed and identified in each sample as a function of the mutation type relative to the reference strain used for analysis. The point mutation differences among the samples highlight the temporal increase in the number of mutations caused by increasing the antibiotic concentrations over the 10 days of the study. The number of mutations observed relative to the untreated (0× FLO) sample induced by the presence of the drug is indicated in parentheses.

Mutation Type		0× FLO	1× FLO	10× FLO	100× FLO	1000× FLO
Complex *	Identified	118	119 (+1)	118	116	116
	All	291	295 (+4)	291	300 (+9)	292 (+1)
Deletion	Identified	19	20 (+1)	21 (+2)	21 (+2)	20 (+1)
	All	39	41 (+2)	42 (+3)	43 (+4)	41 (+2)
Insertion	Identified	3	4 (+1)	3	4 (+1)	3
	All	13	15 (+2)	14 (+1)	15 (+2)	13 (+2)
SNP **	Identified	781	790 (+9)	789 (+8)	779	779
	All	1415	1411	1388	1419 (+4)	1376

* A compound mutation that may involve multiple insertions, deletions and substitutions; ** single-nucleotide polymorphism. FLO—florfenicol.

**Table 6 antibiotics-12-01728-t006:** Mutations affecting genes relevant for antimicrobial resistance were all single-nucleotide polymorphisms (SNPs).

Gene	1	2	3	4	5	Nucleotide Acid Replacement	Effect	Product
*mdtC*	x	x	x	x	x	C-A	missense_variant c.1860C>A p.Ser620Arg	multidrug efflux RND transporter permease subunit *mdtC*
*mdtN*	x	x	x	x	x	G-A	missense_variant c.671C>T p.Thr224Ile	multidrug efflux transporter periplasmic adaptor subunit *mdtN*
*emrR*			x	x		C-A	missense_variant c.129C>A p.Asn43Lys	multidrug efflux transporter *EmrAB* transcriptional repressor *emrR*
*acrB*				x	x	T-A	missense_variant c.1706A>T p.Gln569Leu	efflux RND transporter permease *acrB*
*acrR*					x	G-A	missense_variant c.506G>A p.Gly169Asp	multidrug efflux transporter transcriptional repressor *acrR*
*robA*					x	C-T	missense_variant c.208G>A p.Ala70Thr	MDR efflux pump *acrAB* transcriptional activator *robA*

1—0× FLO; 2—1× FLO; 3—10× FLO; 4—100× FLO; 5—1000× FLO. FLO—florfenicol.

## Data Availability

The datasets used and/or analyzed during the current study are available from the corresponding author on reasonable request. The sequencing files are available at the https://submit.ncbi.nlm.nih.gov/subs/wgs_batch/SUB13967108/overview (accessed on 13 November 2023).

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
