# Peer review of "In Vitro Microevolution and Co-Selection Assessment of Florfenicol Impact on Escherichia coli Resistance Development"

_antibiotics, 2023, doi:10.3390/antibiotics12121728_

Round 1
Reviewer 1 Report
Comments and Suggestions for Authors
The article entitled” In vitro evolution and co-selection assessment of florfenicol impact on Escherichia coli resistance development” tackled an interesting topic. It is a very complex study with some great results. However, I have some concerns that I will address as follows:
1. Line 18 – Abstract – explain the AMEG B abbreviations. Usually, the abstract is the first contact with the paper and should be as clear as possible.
2. Introduction – describe the utility of the MEGA-plate method, the advantages and perhaps some disadvantages. Since this technique is in the center of your experiment, I would encourage you to tackle it more in-depth.
3. Table 1 - Explain why the MIC does not have an ascending/descending trend and vary on different levels of exposure to florfenicol.
4. Figure 2 – Explain the main findings that this figure shows. A simple sentence – line 112/113 – is not enough.
5. Please review the entire article and use italics for all the genes. There are some missing like in the line 282, line 287 etc.
6. Materials and Methods – you used an adapted MEGA-plate. Please add a visual representation/photo/draw of you adaptation and a step-by-step approach to manufacturing it and the usage. This might prove to be an extremely helpful feature for this paper and for future citing.
7. Line 391 - you mentioned based on your experience – Please add the reference that support your statement or explain in detail why the additional capsule is needed.
8. Line 412 - you mentioned values in bold. What values? Please rephrase so it is clear.
9. Explain why you used CLSI and not EUCAST.
0. Conclusions need to be rewrite – Line 512/513 – the explanation of how efflux pumps works is not necessary here (maybe in introduction). Please point out the main findings in an organized and logic way.
Author Response
Dear Reviewer1!
Thank you very much for your very important and useful feedback, which has been fully corrected in the manuscript as follows. The corrected parts are highlighted in green in the manuscript.
- We have corrected the abbreviation of AMEG B (Lines 17-20).
- In the Introduction, we have explained the usefulness and advantages of the MEGA-plate method and highlighted its potential drawbacks, thus examining the methodology from all sides, providing a self-critique (Lines 66-76).
- Explained the trends in MIC values and why they changed with florfenicol (Lines 89-92).
- We have explained the significance of Figure 2, which is a quality certification of the samples - but we do not believe that a more detailed explanation is necessary (Lines 117-120).
- Corrected as requested and all gene names are italicised.
- A photograph (Figure 4) of the plate wilted during the use of the active ingredient florfenicol has been included (Lines 410-412). The adapted methodology simply means that we used a smaller polycarbonate plate, the main steps are explained, so we believe that the original MEGA-plate methodology referred to is not to be described unnecessarily, as the steps of preparation were followed and the deviations are explained (Lines 377-397).
- We have explained as requested, as it was our own experience and therefore not referenced (Lines 397-400).
- Reference was made to Table 1 (Line 427).
- The CLSI standard describes an internationally accepted methodology for both MIC testing and ESBL production testing. Although EUCAST also has such formal testing methodologies, both are equally accepted worldwide. We mainly use CLSI for no particular reason. Otherwise, the two methodological standards typically refer to each other. It is difficult to do justice, which I do not think is our business in their competition with each other.
- Thank you very much for pointing this out. We have restructured the conclusions so that they appear more logical and organised (Lines 517-538).
Reviewer 2 Report
Comments and Suggestions for Authors
Title: In vitro evolution and co-selection assessment of florfenicol impact on Escherichia coli resistance development
This manuscript is well-written by the authors. I do believe that if they can improve the manuscripts following all comments. It might have a chance to publish in the journal.
Comments
1. Topic: Please rewrite. It is complicated. In my opinion, the authors investigated antimicrobial resistance of florfenicol in E. coli. It is not the evolution.
Abstract
2. Please write the scientific name of the bacteria, in vitro, in vivo in italic.
3. The introduction in the abstract should be reduce. One key sentence is enough.
4. Please identify the objective and the methods of this study. It may be better if the author uses passive voice instead of, we examined……
5. It will be better if the authors explain the results by the values or quantitative data.
6. Please modify the keywords.
Results
7. Figureb1: Please prepare the figure for the scientific data. Some symbols in this figure look like a cartoon.
8. Please increase the resolution of the figures. Some sentences in the figures are not clear.
9. For the names of genes should be written in italic.
10. Line 137-138 should be re-written.
Discussion
11. Please add the information of the discussion. Try to compare the results (the author’s hypothesis) with other finding by other researchers.
12. Please delete some introduction sentences in the discussion part.
Materials and methods
13. Line 362: E. coli should be written in italic.
14. 361-367: The bacterial culture condition of E. coli should be described.
15. Do the authors perform statistical analysis?
Conclusion
16. Please modify the conclusion. It should be summarized on the key finding. Please don’t repeat the results and discussion.
References
17. The references of 2022,2021, and 2023 are suggested to be cited.
18. Please remove some old references.
Comments on the Quality of English LanguageSome sentences in the manuscript should be edited the grammar.
Author Response
Dear Reviewer2!
Thank you very much for your useful and good insights, which will help us to improve this very important research topic. We have taken your advice and have tried to correct all the suggestions in the manuscript, taking them into account as much as possible. Changes have been highlighted in yellow.
Title 1: The subject is indeed complex, but the methodology was designed specifically for evolutionary studies such as the one we have carried out. Although the size of the system is limited, the estimated several generation changes in E. coli make the system suitable for evolutionary studies, we believe, due to the continuous selection pressure. It may indeed be more appropriate to speak of microevolutions, as we do not wish to completely take them out of the manuscript and have therefore clarified the title of the manuscript to this effect (Line 2).
2 Thank you very much for pointing this out, the name Escherichia coli appears in italics in our view (Line 21), we have corrected the term in vitro to italics (Line 37), we have not included the term in vivo in the manuscript.
- Thank you very much for your suggestion to reduce, however the other reviewer suggested exactly the opposite (highlighted in green), we cannot do justice between the two conflicting suggestions, please discuss this between the two reviewers or with the editor.
- Thank you very much, as requested we have rewritten the parts in question in a suffering structure (Line 20, Lines 23-24). We have concisely defined our study objectives (Lines 21-23).
- Thank you, we have introduced quantitative data as requested (Lines 26-27).
- We have also corrected the objectionable word in the keywords as requested in the title (Line 39).
- Thank you very much for the scientific suggestion, we have deleted the non-scientific text bubble (Figure 1) that was objected to (Line 121).
- The figures are very high resolution. Please do not try to view and interpret them in the manuscript, but in the attached files. The enlargable figures inserted in the manuscript tend to work perfectly well in the published manuscript, which I have observed in several manuscripts of this type and we have relied on; we did not want to insert five separate figures instead of one.
- Thank you very much for pointing this out, we have corrected it to italics in all cases.
- As requested, we have reworded the offending sentence, hopefully to give a nicer context to what it says (Lines 158-159).
- Thank you very much for bringing this to our attention. In preparing the manuscript, we found few comparable or similar studies that examined things in this context. The literature is much more favourable for genes, but the literature on comparisons in the light of phenotypic studies is poor. We have tried to include studies that are related at some level (Lines 278-280).
- As requested, we have deleted a sentence from the beginning of the Discussion.
- Thank you for pointing out the lack of italics, which we have corrected (Line 370).
- We have accurately described the culture conditions of the E. coli bacteria (Lines 404-408).
- Since we worked with one sample in each case in this study, no statistical test could be performed.
- Thank you very much for your comments, the Conclusion is fully restructured as requested by the other reviewer, with the most important findings from this point of view highlighted in green (Lines 517-538).
- Thank you very much for the suggestion to use as recent publications as possible. Where possible, we have replaced them with more recent literature (5, 11, 16, 17, 19, 25, 36, 37).
- Old references have been reduced as requested.
Round 2
Reviewer 1 Report
Comments and Suggestions for Authors
I accept in current form
Author Response
Dear Reviewer 1,
Thank you very much.
Yours sincerely,
Adam Kerek
Reviewer 2 Report
Comments and Suggestions for Authors
-This manuscript is well-written by the authors. It can have a chance to publish in the journal.
-By the way, the word "In vitro" (at the title) should be written in italic.
Comments on the Quality of English Language-The authors have edited the grammar.
Author Response
Dear Reviewer 2,
Thank you very much.
We have corrected the term "in vitro" and it is now written in italics.
Yours sincerely,
Adam Kerek